# Anti-Corrosion Coating Formation by a Biopolymeric Extract of *Artemisia herba-alba* Plant: Experimental and Theoretical Investigations

Walid Daoudi [1,*,†], Abdelmalik El Aatiaoui [1], Omar Dagdag [2], Kaoutar Zaidi [3], Rajesh Haldhar [4,†], Seong-Cheol Kim [4,*], Abdelouahad Oussaid [1], Abdelouahad Aouinti [3], Avni Berisha [5], Fouad Benhiba [6], Eno Effiong Ebenso [2,*] and Adyl Oussaid [1]

1 Laboratory of Molecular Chemistry, Materials and Environment (LCM2E), Departement of Chemistry, Multidisciplinary Faculty of Nador, University Mohamed I, Nador 60700, Morocco
2 Centre for Materials Science, College of Science, Engineering and Technology, University of South Africa, Johannesburg 1710, South Africa
3 Laboratory of Applied Chemistry & Environment (LCAE), Department of Chemistry, Faculty of Sciences, Mohammed First University, Oujda 60000, Morocco
4 School of Chemical Engineering, Yeungnam University, Gyeongsan 38541, Republic of Korea
5 Department of Chemistry, Faculty of Natural and Mathematics Science, University of Prishtina, 10000 Prishtina, Kosovo
6 Laboratory of Advanced Materials and Process Engineering, Faculty of Science, Ibn Tofail University, Kenitra 14000, Morocco
* Correspondence: walid.daoudi@ump.ac.ma (W.D.); sckim07@ynu.ac.kr (S.-C.K.); ebensee@unisa.ac.za (E.E.E.)
† These authors contributed equally to this work.

**Abstract:** In this modest work, a local biopolymer (CHA), biodegradable, non-toxic, and soluble in acidic media, was extracted from the plant *Artemisia herba-alba* located in the eastern region of Morocco, and characterized by FT-IR, in order to valorize it as a corrosion inhibitor of mild steel in 1 M HCl medium. The electrochemical tests show that the extract is an excellent corrosion protective agent, reaching a maximum value of 96.17% at the concentration of 800 mg/L in the inhibitor. The potentiodynamic polarization (PDP) curves indicate the mixed behavior of the extract, to reduce the current density from 3.445 mA/cm$^2$ to 0.104 mA/cm$^2$ in the presence of 800 mg/L in the inhibitor. The biopolymer CHA of the extract of *Artemisia herba-alba* undergoes the Langmuir adsorption isotherm, whose adsorption energy is −20.75 kJ/mol, which is attributed to the presence of electrostatic and covalent bonds. In addition, the visualization of the metal surface by a scanning electron microscope (SEM) indicates the formation of a protective layer formed by the extracts of *Artemisia herba-alba*, which confirms the protective characteristic of the extract used. Theoretical investigations by DFT, MD, and MC confirm previous experimental results.

**Keywords:** *Artemisia herba-alba*; FT-IR; corrosion; PDP; SEM; DFT

## 1. Introduction

Generally, most metals and their alloys such as steel, copper, aluminum, and zinc are widely used in various industrial sectors due to their mechanical and physical properties, as well as their availability in the industrial market and their cost-effectiveness [1,2]. In addition, acid solutions, especially 1 M HCl, are used in various industrial sectors, such as industrial cleaning, oil well acidification, and many industrial synthesis methods [3]. Despite the global problem of increasing erosion of metals, there are still issues to be considered when exposing these materials at industrial levels, which leads to global economic depletion [4]. Nevertheless, various approaches used to reduce and prevent corrosion of mild steel under the conditions of industrial operations, including biodegradable extracts, are often proposed as the most economical and non-toxic option [5–9].

Recently, the majority of the research has been oriented toward the use of extracts from aromatic plants as protective agents against the corrosion of metals in industrial operative conditions, because of their availability and their non-toxicity. At this stage, the vegetative growth of the plant *Artemisia herba-alba*, in the world, particularly in Morocco, makes this plant important at the industrial level. Indeed, the constituents of *Artemisia herba-alba* are widely used as pharmaceutical additives, nutraceuticals, and toxicological and anticorrosive agents. Boudalia et al. [10] extracted the essential oil of the plant *Artemisia herba-alba* and used it as a corrosion inhibitor of stainless steel in phosphoric acid. Electrochemical tests showed that the oil is a good corrosion inhibitor that reaches an optimal value of 88% at 1 g/L in oil. Radjai et al. [11] indicated that the methanolic extract of the stems and leaves of the plant *Artemisia herba-alba* is an ecological corrosion inhibitor of mild steel in 1 M HCl medium, whose effectiveness increases with the increase in the concentration of the inhibitor. Similarly, Echihi et al. [12] studied the inhibitory effect of the methanolic extract of the plant *Artemisia herba-alba* to protect mild steel against the attack of $H^+$ and $Cl^-$ ions, in which the electrochemical tests (EIS) lead to the inhibitory protection of about 90% at 500 ppm, whose adsorption is performed according to the Langmuir isotherm. On the other hand, Bouyanzer et al. [13] valorized the oil from the plant *Artemisia herba-alba*, from the eastern region of Morocco, as a corrosion inhibitor. Hechiche et al. [14] examined the adsorption of the essential oil of *Artemisia herba-alba*, on aluminum in 1 M HCl medium, whose inhibitory efficiency passes 91% in 3 g/L at the temperature of 333 K. Boumhara et al. [15] studied the inhibitory effect of the essential oil of *Artemisia herba-alba*, on mild steel in 0.5 M $H_2SO_4$. An inhibitory efficiency that reaches 87% was recorded at the concentration of 2.76 g/L by gravimetric and electrochemical tests, whose inhibitory efficiency decreases with the increase in temperature.

The novelty of this research work is to extract a biopolymer "cellulose" (CAH) from the waste of the *Artemisia herba-alba* plant and use it as a mild steel corrosion inhibitor in the chemical engineering industry. However, cellulose is the main constituent of the cell wall and is considered the most industrially important organic polymer. Recently, the extraction of cellulose from the plant *Artemisia herba-alba* has become a subject of interest thanks to their availability in nature. In this work, we extracted the biopolymer CHA from the *Artemisia herba-alba* plant and characterized it by FT-IR, in order to compare it with the commercial one. Gravimetric, electrochemical, and transient tests were used to assess inhibitory efficacy. In addition, the metal surface was visualized by SEM coupled to EDX to show the adsorption of the extract on the substrate. Finally, calculations by density functional theory (DFT), simulations by molecular dynamics (MD), and simulations by the Monte Carlo (MC) method were carried out in order to study the geometry of polymer structures and to find the existence of a correlation between the molecular structure of the inhibitor studied and their inhibitory potency. This has allowed quantum chemistry to become an indispensable complement to experimental studies.

## 2. Materials and Methods

### 2.1. Materials

The plant *Artemisia herba-alba* used in this work was recovered from the Eastern region of Morocco. The stems and leaves of *Artemisia herba-alba* were mixed in a blender with rotating blades. The stems underwent dry grinding for 5 min until a powder was obtained. The extractable amount was the only fraction that can be isolated without being degraded or modified. To extract the extractives, 15 g of the powder of the stems of the recovered *Artemisia herba-alba* waste was introduced into a 250 mL flask containing distilled water (V = 100 mL), THF (V = 45 mL), and EtOH (V = 45 mL); the mixture was heated under reflux for 5 h at a temperature of 100 °C and under stirring. After filtration of the solvent, the powder was washed three times with ethanol alone and put to dry at room temperature until weight stabilization. The powder was treated with a volume V = 120 mL of 1 M NaOH solution, and the mixture was heated under reflux for 2 h at 100 °C. The suspension was filtered and washed thoroughly twice with water and three times with ethanol. The soda

eliminates the hemicelluloses and causes swelling of the fibers. The latter is transformed into individualized microfibers after bleaching with NaClO₂ (Scheme 1) [16].

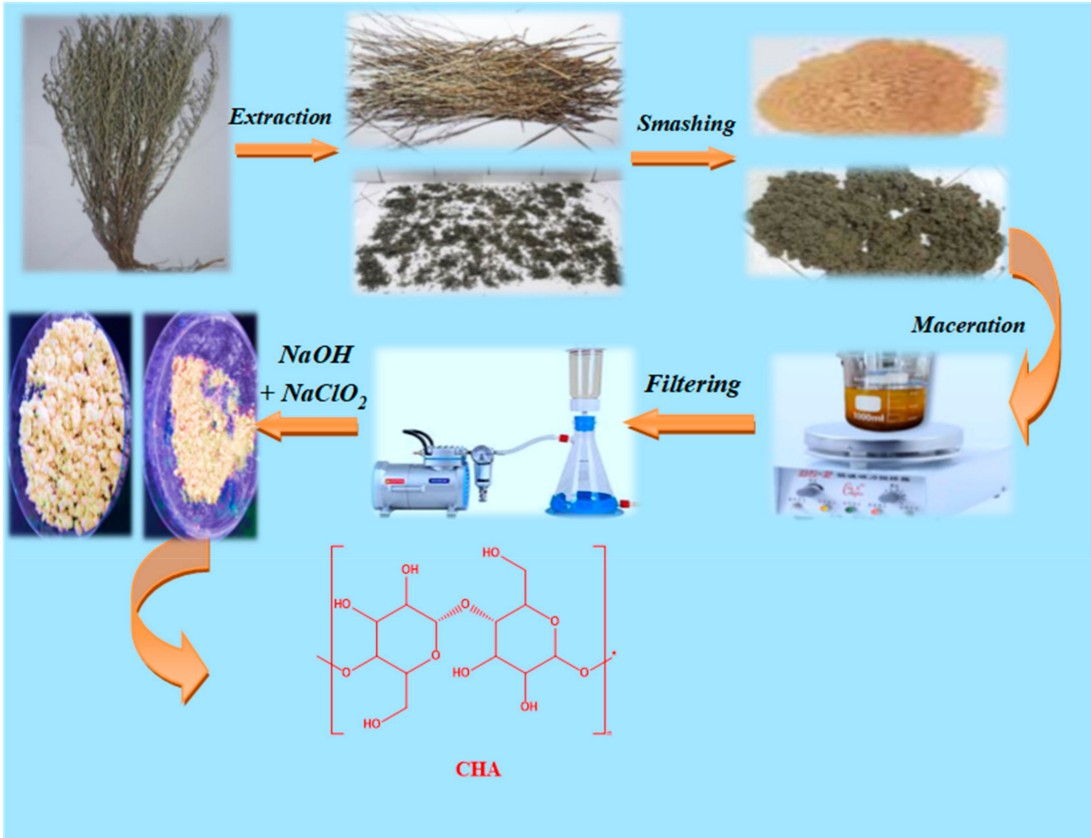

**Scheme 1.** Diagrammatic illustration of CHA extraction.

The mild steel employed in this project had a chemical composition of 0.06 S, 0.04 O, 0.60 Mn, 0.31 Si, and 0.22 C, and the other components were iron. Moreover, the commercialized product HCl 37% (Sigma Aldrich, Saint Louis, MO, USA), with which the concentration of CHA is somewhere between 200 mg/L and 800 mg/L, was used to make the 1 M HCl acid solution.

*2.2. Gravimetric Tests*

The mass loss measurement technique has the advantage of being simple to implement and does not require significant equipment [17]. It consists in weighing with a precision balance, with the difference in mass undergone by a surface sample during 6 h of immersion in a corrosive solution inhibited and not inhibited by CHA at the temperature of 298 K. Before each test, the metal surface was polished with increasing fine-grit paper: 1200-1000-600-220-180-100, to remove pitting and defects, and before emerging into the corrosive medium, it was rinsed with distilled water and dried with hot air. The corrosion rate $C_R$ and the inhibitory efficiency $\mu_w$ are given by the following relations, respectively [18]:

$$C_R = \frac{m_i - m_f}{A \times t} \tag{1}$$

$$\mu_w\ (\%) = \frac{C_R^0 - C_R^i}{R_R^0} \times 100 \tag{2}$$

where $C_R^0$ and $C_R^i$ are the corrosion rates without and with CHA extract, respectively. A is the surface area of the mild steel (MS).

*2.3. Electrochemical Tests*

The three-electrode potentiostat model PGZ101 (Paris, France) used for the electrochemical experiments was operated by the voltameter 4 program. Mild steel served as the working electrode. The PDP potential-dynamic polarization curves ranged from $-800$ through $-200$ mV at 298 K with a polarization sweep rate of around 1 mV·s$^{-1}$ after the mild steel had been submerged in the aggressive solution for 30 min. The Nyquist and Bode plots, on the other hand, were named in accordance with an alternating signal including an amplitude of 10 mV and a frequency range of 100 KHz to 10 mHz.

*2.4. UV-Visible Test*

To examine the possibility of iron–inhibitor complex formation, we tested solutions inhibited with 800 mg/L of CHA biopolymer before and after immersion of the steel in the solution using quartz cells and a Shimadzu$^{TM}$ UV-1800 (Markham, ON, Canada) dual beam spectrophotometer.

*2.5. Visualization of Metal Surface*

After polishing the metal surfaces, the iron coupons were immersed in the acid solutions in the absence and presence of 800 mg/L of CHA at 298 K for 24 h. The morphology of the metal surface was visualized by a scanning electron microscope SEM with a magnification of about 50 μm.

*2.6. Computational Studies*

2.6.1. DFT

The DFT calculations were performed using the Dmol3 module incorporated through the Biovia Materials studio program (version 20.1.0.2728, BIOVIA Materials Studio 2020, Dassault Systèmes, Vélizy-Villacoublay, France) [19,20]. Geometry optimizations were carried out using the Generalized Gradient Approximation [21], which makes use of the M06l [22,23] along with the double numerical basis set including polarization (DND) [24]. Aqueous-phase DFT studies using the Conductor-like Screening Model (COS-MO) were conducted [25].

2.6.2. MC and MD Simulations

Using 9 layers of Fe atoms, it is possible to model the interactions among steel surfaces as well as CHA biopolymer inhibitors inside the simulated aggressive environment. The slab model had these three dimensions: 22.341 Å × 22.341 Å × 16.214 Å + 35 Å vacuum layer at the C axis, and there was a vacuum layer with the following composition: 10 chlorides, 10 hydroniums, 850 water molecules, and 1 inhibitor. Before the MD stage, the shape of the simulation boxes was optimized using the Forcite module built into the Biovia software package (version 20.1.0.2728, BIOVIA Materials Studio 2020, Dassault Systèmes, Vélizy-Villacoublay, France) (energy convergence tolerance of 0.001 kcal/mol; atom-based summing approach was used with a cutoff distance of 12.5 Å, spline width of 1 Å, and buffer of 0.5 Å).

The fixed-volume/fixed-temperature (NVT) canonical ensemble was used for MD at 298 K [26–28] with a simulation duration of 1000 ps (1 fs time step) [29–31]. The Berendsen thermostat makes sure that the temperature is kept under control. The MC and MD of the system in the condensed phase were calculated using the COMPASSIII (BIOVIA Materials Studio 2020, Dassault Systèmes BIOVIA, Cambridge, UK) forcefield [32–34]. The complete MD trajectory was examined using the Radial Distribution Function (RDF) method [35].

## 3. Results and Discussion

*3.1. Confirmation of the Chemical Structure of Extracted CHA*

To identify the phytochemical functional groups, present in the CHA biopolymer, we used a reliable technique, FT-IR spectroscopy, in order to characterize our recovered

bio-organic material. Figure 1 shows the FT-IR spectrum of commercial cellulose and that of cellulose extracted from the stems and leaves of the plant Artemisia herba-alba.

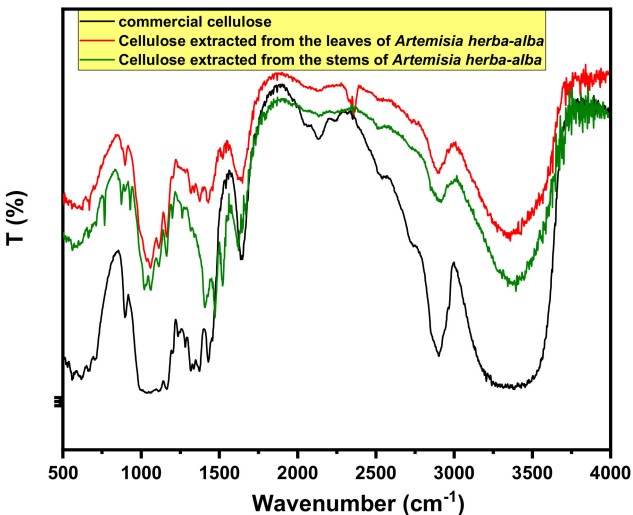

**Figure 1.** FT-IR spectra of the biopolymer CHA.

Comparing the two FT-IR spectra, that of the commercial cellulose and that of the biopolymer CHA extracted from the stems and leaves of Artemisia herba-alba, we can demonstrate that these spectra are superimposable and that the white solid found at the end of the isolation process used on these stems and leaves correspond to the biopolymer. By analyzing the spectrum of the extract, we can attribute some present bands to the corresponding vibrators. Indeed, between 3000 and 3600 $cm^{-1}$, we find the bands of vibrations of elongations of the bonds O–H of the primary and secondary alcohol functions. The vibration of deformation in the plane of function O–H in C6 is at 1275 $cm^{-1}$. The vibration band around 2900 $cm^{-1}$ is attributed to the elongation vibrations of the C–H bond. The in-plane deformation vibrations of $CH_2$ in C6 (shear, agitation) are at 142 and 1325 $cm^{-1}$. The antisymmetric elongation vibration of the β-glycosidic C–O–C bond appears at 1163 $cm^{-1}$. This band is used later to normalize the spectra as it is expected to undergo no change during the modification steps. Water adsorbed on cellulose appears at 1639 $cm^{-1}$. However, the absence of a vibration band around 1775 $cm^{-1}$, which corresponds to the elongation vibration of the C=O bond, confirms the absence of lignin residues on cellulose chains, which shows the total disappearance of all organic encrustations of the wall (lignin) during the bleaching step.

### 3.2. Gravimetric Test

### 3.2.1. Effect of CHA Extract Concentration

The gravimetric technique is more frequently used in the field of corrosion, due to its high accuracy, ease of execution, and cost effectiveness, whose purpose is to examine the influence of CHA content on the corrosion of mild steel in 1 M HCl. Figure 2 shows the evolution of the inhibitory efficiency and the corrosion rate of the biopolymer CHA extract at the temperature of 298 K.

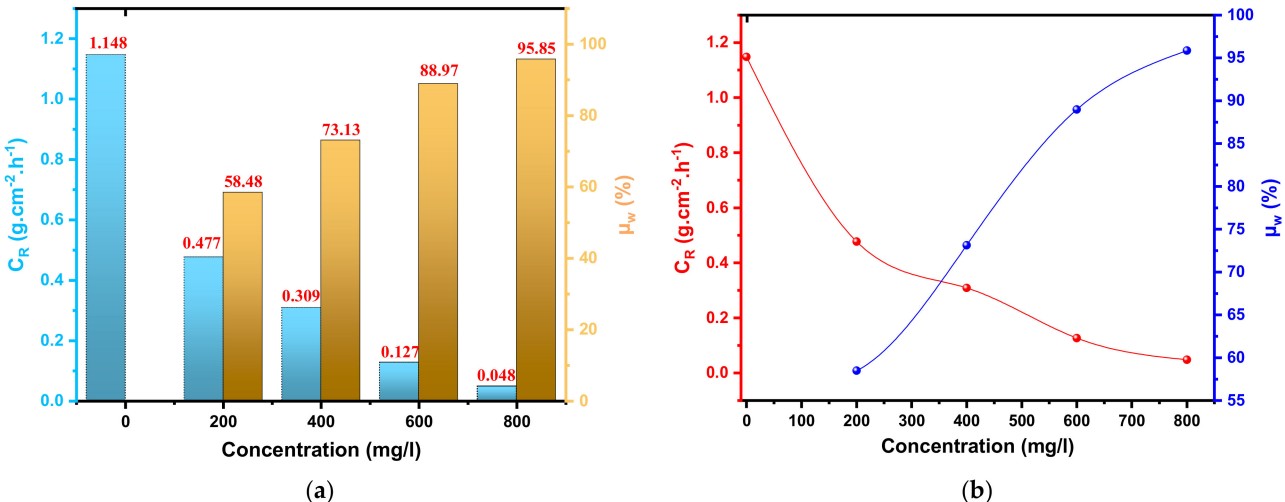

**Figure 2.** The evolution of inhibitory efficiency and corrosion rate in 1 M HCl without and with *Artemisia herba-alba* biopolymer CHA (**a**,**b**).

Based on the results of Figure 2, we observe a remarkable reduction in the corrosion rate when adding the CHA extract to the aggressive 1 M HCl solution. Indeed, the decrease in the corrosion rate increases with the increase in the molar content of the added CHA extract. Moreover, an excellent inhibitory efficiency is recorded at the concentration of 800 mg/L of CHA, whose optimal efficiency value is about 95.85%. Gravimetric results indicate that the biopolymer adsorbs easily on the metal surface due to the presence of several active sites on the cellulose such as O oxygen atoms and $\pi$-bonds, which subsequently facilitates the chemical adsorption of the heteroatom binding doublets on the iron active sites [36].

### 3.2.2. Effect of Temperature and Activation Kinetics Parameters

Temperature is a kinetic factor that can influence the inhibitory behavior of the inhibitor and the nature of the metal–inhibitor interaction [37]. Gravimetric tests were performed with and without CHA at different temperatures (ranging from 308 to 328 K) for one hour and at the maximum concentration of 800 mg/L in CHA. The experimental data obtained are presented in Table 1.

**Table 1.** The values of the corrosion rate and inhibitory efficiencies at different temperatures, as well as the kinetic parameters of activation.

| Inhibitor | 308 K | | 318 K | | 328 K | | $E_{act}^0$ (kJ/mol) | $\Delta H_{act}^0$ (kJ/mol) | $\Delta S_{act}^0$ (J mol$^{-1}$ K$^{-1}$) |
|---|---|---|---|---|---|---|---|---|---|
| | $C_R$ (g/cm$^2$·h) | $\mu_w$ (%) | $C_R$ (g/cm$^2$·h) | $\mu_w$ (%) | $C_R$ (g/cm$^2$·h) | $\mu_w$ (%) | | | |
| HCl | 1.426 | - | 2.610 | - | 3.983 | - | 43.20 | 40.56 | −110.75 |
| CHA | 0.126 | 91.16 | 0.313 | 88.01 | 0.804 | 79.81 | 77.79 | 75.15 | −18.95 |

The data in Table 1 show that the corrosion rate increases with increasing temperature in the whole temperature range studied so that the metal protection by CHA decreases slightly with increasing temperature. This can be explained by the positive contribution to the corrosion phenomenon, which promotes the rapid desorption of the CHA biopolymer at the metal surface. Furthermore, based on the experimental data obtained from the temperature effect, the kinetics activation parameters such as activation energy $E_{act}^o$, enthalpy $\Delta H_{act}^o$, and entropy $\Delta S_{act}^o$ are determined according to the following Arrhenius and Arrhenius transition equation:

$$\ln(C_R) = \ln(K) - \frac{E_{act}^0}{RT} \qquad (3)$$

$$\ln\left(\frac{C_R}{T}\right) = \left[\ln\left(\frac{R}{Nh}\right) + \left(\frac{\Delta S_{act}^0}{R}\right)\right] - \frac{\Delta H_{act}^0}{RT} \qquad (4)$$

where $R$ is a constant of perfect gases, $K$ is the pre-exponential factor, $N$ is Avogadro's number, and h is Planck's constant. Figure 3 represents the evolution of ln(CR) and ln(CR/T) as a function of 1000/T in the absence and presence of 800 mg/L of CHA.

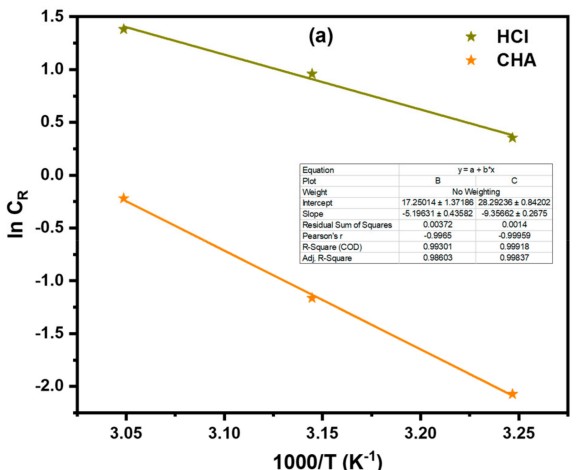 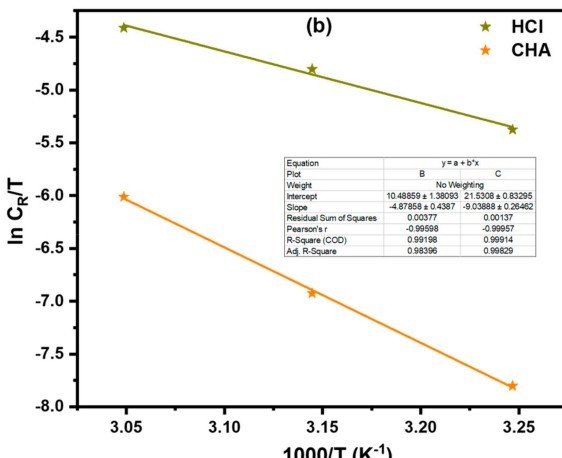

**Figure 3.** Arrhenius (**a**) and Arrhenius transition (**b**) curves for the corrosion rate of mild steel coupons immersed in the acid solution without and with the extract CHA of *Artemisia herba-alba*.

The linear regression coefficient $R^2$ for all plots was close to 1, indicating that the corrosion of the mild steel in the acidic environment can be elucidated using the kinetic model. However, it is clearly noted that the activation energy value $E_{act}^0$ for the inhibited solution is higher than that of the uninhibited solution. This implies the elevation of the energy barrier of the corrosion reaction, and it means that the corrosion reaction will be more difficult due to the adhesion of this inhibitor to the metal surface, forming a protective layer, and consequently resulting in a lower dissolution of iron [38]. Analysis of the values $\Delta H_{act}^0$ shows that the value of this kinetic magnitude is higher in the presence of the CHA extract, suggesting that this dissolution is slow in the presence of the CHA inhibitor [39]. In addition, the activation entropy $\Delta S_{act}^0$ becomes larger in the presence of 800 mg/L of CHA compared to the blank, which means that there is an increase in disorder during the transformation of iron oxides into the iron–inhibitor complex [40].

### *3.3. Electrochemical Tests*

### 3.3.1. Potentiodynamic Polarization Curves (PDP)

The exploitation of the potentiodynamic polarization curves (PDP) allows for acquiring the inhibiting efficiencies of the CHA extract, in order to exhibit the behavior of the studied inhibitor. Figure 4 shows the PDP graphs of mild steel corrosion without and with different concentrations of CHA at 298 K.

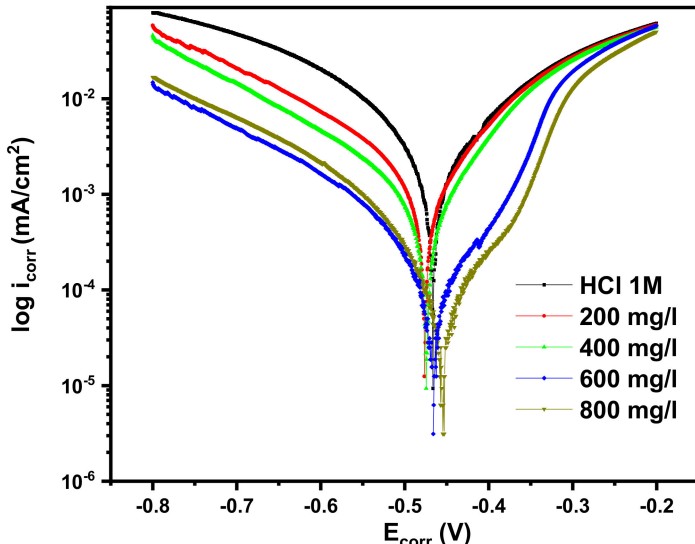

**Figure 4.** Graphical PDP of corrosion of mild steel (MS) in the absence and presence of *Artemisia herba-alba* extract CHA.

The different values of electrochemical parameters extracted from PDP graphs such as corrosion current density $i_{corr}$, corrosion potential $E_{corr}$, cathodic βc and anodic βa Tafel slopes, and inhibition efficiency $\mu_{PDP}$ for different concentrations of CHA extract in the aggressive environment are reported in Table 2. The inhibitory efficiency calculated from PDP is defined as follows:

$$\mu_{PDP}(\%) = \frac{i^0_{corr} - i^i_{corr}}{i^0_{corr}} \times 100 \tag{5}$$

**Table 2.** Electrochemical parameter values extracted from PDP plots in the presence and absence of various tested concentrations of CHA extract at temperature of 298 K.

| Inh | Conc. | $E_{corr}$ (mV/SCE) | $i_{corr}$ (mA/cm$^2$) | Tafel's Slopes | | $\mu_{PDP}$ (%) |
|---|---|---|---|---|---|---|
| | | | | $\beta_a$ (mV) | $-\beta_c$ (mV) | |
| HCl | 1 M | −466.39 | 3.445 | 184.40 | 179.90 | - |
| CHA | 200 mg/L | −476.00 | 1.502 | 140.90 | 176.20 | 56.40 |
| | 400 mg/L | −474.26 | 1.095 | 128.40 | 200.60 | 68.21 |
| | 600 mg/L | −480.01 | 0.339 | 109.60 | 187.60 | 90.16 |
| | 800 mg/L | −454.31 | 0.104 | 90.40 | 144.80 | 96.98 |

Figure 4 shows that the shapes of potentiodynamic polarization graphs remain unchanged in the presence of different concentrations of CHA extract compared to that of the blank, which is accompanied by a decrease in current density from 3.445 mA/cm$^2$ to 0.104 mA/cm$^2$ in the absence and presence of 800 mg/L of CHA, respectively. The maximum inhibitory efficiency is about 96.98%, which shows the excellent inhibitory performance of the polymeric extract of the plant *Artemisia herba-alba*. Interestingly, the electrochemical results reveal a remarkable change in the anodic βa and cathodic βc Tafel's law when more and more content of the CHA extract is added, suggesting that the biopolymer effectively retards the iron oxidation and hydrogen ion reduction [41]. Furthermore, Tafel's law is well verified and the corrosion potential $E_{corr}$ is not affected either by the addition of the CHA inhibitor to the blank or by the concentration effect of each CHA. Therefore, it can be concluded that the CHA extract has a mixed characteristic [42] and that the corrosive system is controlled by an activation, in which the addition of an inhibitor does not affect the mechanism of the hydrogen evolution reaction [43].

### 3.3.2. Electrochemical Impedance Spectroscopy

To identify the complex mechanisms, which include many reaction steps and have different kinetic properties, the Electrochemical Impedance Spectroscopy (EIS) technique is performed, in order to understand the reaction mechanism that will take place at the metal–solution interface. The EIS measurements of mild steel corrosion without and with CHA extract in the Nyquist and Bode diagrams are shown in Figure 5.

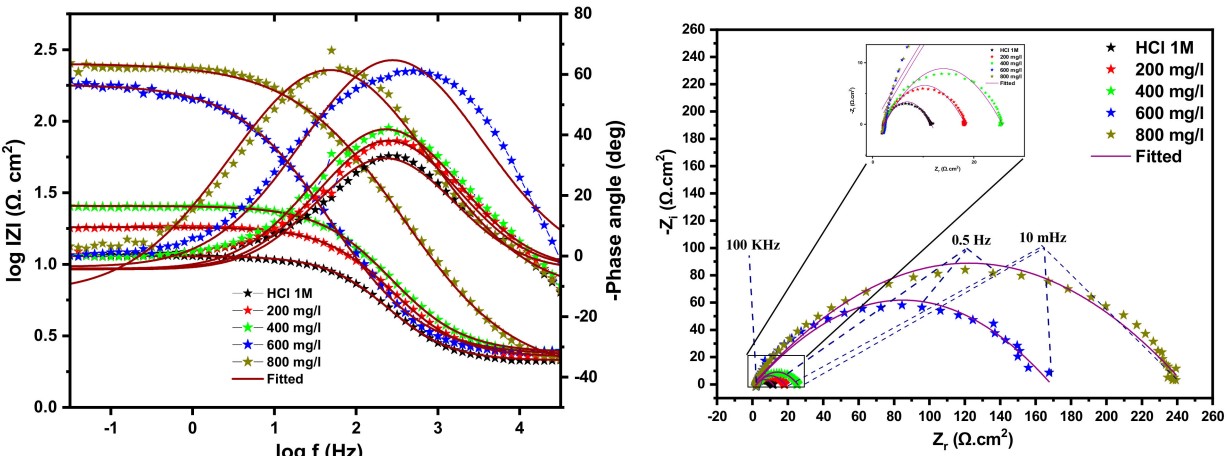

**Figure 5.** Nyquist and Bode plots in the presence and absence of the CHA extract at temperature 298 K.

The EIS plots in the Nyquist diagram show the presence of a single capacitive loop in both CHA-inhibited and uninhibited solutions, indicating that the adsorption process is controlled by a charge transfer process [44]. Moreover, the diameter of the loops increases with the increase in the CHA extract content, suggesting the increase in metal protection against the attack of aggressive ions. In addition, in the Bode diagram, we observe the presence of a single time constant, indicating the formation of a double layer on the metal surface [45]. In addition, the metal–solution interface is modeled by an electrical circuit, consisting of a solution resistor $R_S$, connected in series with a constant-phase element (CPE), in parallel with a charge transfer resistor $R_{ct}$, whose purpose is to estimate different electrochemical parameters such as the double-layer capacitance $C_{dl}$, the relaxation time $\tau$, and the inhibitory efficiency $\mu_{SIE}$ according to the following relations, respectively:

$$C_{dl} = \left( Q \times R_{ct}^{1-n} \right)^{1/n} \tag{6}$$

$$\tau = R_{ct} \times C_{dl} \tag{7}$$

$$\mu_{EIS} \ (\%) = \frac{R_{ct}^{i} - R_{ct}^{0}}{R_{ct}^{i}} \times 100 \tag{8}$$

where $R_{ct}^{i}$ and $R_{ct}^{o}$ are the charge transfer resistances in the presence and absence of the inhibitor, respectively. The different electrochemical values are grouped in Table 3.

**Table 3.** Different values of electrochemical parameters extracted from EIS Nyquist curves without and with different molar content of CHA extract at 298 K.

| Inh. | Conc. | $R_s$ ($\Omega \cdot cm^2$) | $R_{ct}$ ($\Omega \cdot cm^2$) | n | $Q$ (mF·S$^{n-1}$) | $C_{dl}$ ($\mu F/cm^2$) | $\tau$ (s) | $\chi^2 \times 10^{-3}$ | $\mu_{EIS}$ (%) | $\theta$ |
|---|---|---|---|---|---|---|---|---|---|---|
| HCl | 1 M | 2.07 | 9.34 | 0.809 | 0.631 | 198.1 | 0.0018 | 1.82 | - | - |
| CHA | 200 mg/L | 2.21 | 16.06 | 0.796 | 0.459 | 131.2 | 0.0021 | 0.4 | 41.48 | 0.4148 |
|  | 400 mg/L | 2.39 | 23.11 | 0.808 | 0.394 | 112.1 | 0.0026 | 1.14 | 59.58 | 0.5958 |
|  | 600 mg/L | 2.37 | 166 | 0.813 | 0.096 | 40.76 | 0.0068 | 1.32 | 94.37 | 0.9437 |
|  | 800 mg/L | 1.99 | 252.6 | 0.810 | 0.095 | 38.97 | 0.0098 | 1.66 | 96.17 | 0.9617 |

From the data in Table 3, it is clearly noted that the values of the $\chi^2$ parameter are very low on the order of $10^{-3}$, which explains the agreement of the experimental impedance spectra with the proposed equivalent electrical circuit. Moreover, it is noticed that the surface heterogeneity parameter n changes with the addition of the CHA extract, indicating the surface modification when the extract CHA of the plant *Artemisia herba-alba* is added [5]. On the other hand, the experimental data show a maximum inhibitory efficiency of about 96.17%, at the concentration of 800 mg/L in CHA extract. In addition, a lowering of the values of the capacity of the double-layer $C_{dl}$ can be explained by the following Helmholtz model:

$$C_{dl} = \frac{\varepsilon^0 \varepsilon}{d} \tag{9}$$

where $\varepsilon^0$ and $\varepsilon$ are the constants of void permittivity and double layer, respectively. $d$ is the thickness of the double layer. The decrease in $C_{dl}$ can be interpreted by the replacement of the water molecules with the cellulose macromolecules so that the layer becomes denser [46]. In addition, it is observed that the relaxation time increases more and more with the increase in the concentration of the CHA extract, which attributes to the spontaneous adsorption of the inhibitor on the metal surface.

On the other hand, in a comparative study of the CHA biopolymer extracted from the plant Artemisia herba-alba, we note, according to Table 4, that the CHA inhibitor presents an excellent inhibitory performance in comparison with the other extracts of different and even low concentrations, which implies the importance of CHA in the industrial sector.

**Table 4.** Comparison of CHA inhibition efficiencies with other extracts.

| Green Inhibitor | Medium and Substrate | Concentration | $\mu_{EIS}$ (%) | Ref. |
|---|---|---|---|---|
| Clinopodium acinos | Mild Steel in HCl | 300 ppm | 92.2 | [47] |
| Psidium guajava linn leaves | Copper in $H_2SO_4$ | 600 mg/L | 92.5 | [48] |
| Beta vulgaris | Mild Steel in HCl | 4 g/L | 85.77 | [18] |
| Achillea Millefolium | Mild Steel in NaCl | 700 ppm | 91.23 | [49] |
| Psidium guajava linn leaves | Copper in $H_2SO_4$ | 600 mg/L | 92.5 | [49] |
| Swertia chirata | Mild Steel in $H_2SO_4$ | 500 mg/L | 92.32 | [50] |
| Akebia trifoliate koiaz peels | Mild Steel in HCl | 800 mg/L | 90 | [5] |
| Artemisia herba-alba | Mild Steel in HCl | 800 mg/L | 96.17 | In this work |

### 3.4. Adsorption Isotherms

To acquire additional information about the biopolymer CHA extracted from the plant *Artemisia herba-alba*, we tested the Langmuir adsorption isotherm. The metal surface coverage rate θ calculated from the potentiodynamic polarization curves PDP is employed to examine the adsorption isotherm based on the correlation coefficient $R^2$ according to the following equation [51]:

$$\frac{C_{ext}}{\theta} = \frac{1}{K_{ads}} + C_{ext} \tag{10}$$

where $K_{ads}$ is the adsorption constant.

Figure 6 shows the plots of the tested adsorption isotherms.

According to the correlation coefficient $R^2$, which is close to unity, it can be said that the adsorption of the biopolymer on the metal surface obeys the Langmuir isotherm. This type of isotherm implies the assumption of no interaction between the adsorbent and the adsorbate, whose $K_{ads}$ are related to the adsorption energy $\Delta G_{ads}$ by the following equation [52]:

$$K_{ads} = \frac{1}{1000} \exp\left(\frac{-\Delta G_{ads}}{RT}\right) \tag{11}$$

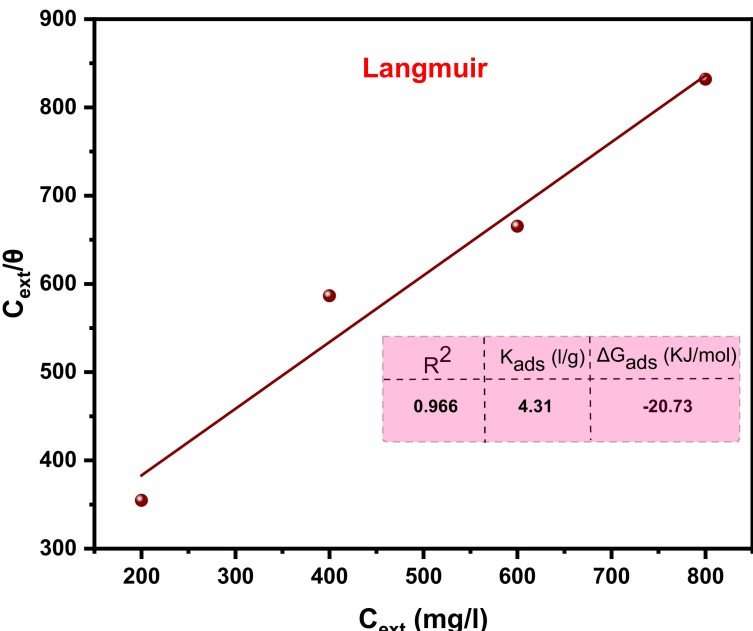

**Figure 6.** Langmuir adsorption isotherm of the extract CHA of the plant *Artemisia herba-alba*.

According to the negative sign of adsorption energy $\Delta G_{ads}$, it indicates the spontaneity of the adsorption of CHA extract on the metal surface. In general, it is known that if the value is less than $-20$ KJ/mol, it implies the presence of electrostatic interactions between the metal dipoles and those of the inhibitor. On the other hand, if $\Delta G_{ads}$ is greater than or equal to $-40$ KJ/mol, chemical bonds between the inhibitor heteroatoms and the vacant iron doublets are formed [53]. In this case, the value of $\Delta G_{ads}$ is between $-20$ kJ/mol and $-40$ kJ/mol, suggesting that a mixed behavior of the CHA extract takes place with the predominance of weak bond formations during metal inhibition.

*3.5. UV-Visible Test*

The UV-visible technique was performed to evaluate the possibility of the formation of an iron-CHA complex. Figure 7 shows the UV-visible plots of the CHA extract before and after immersion of the mild steel.

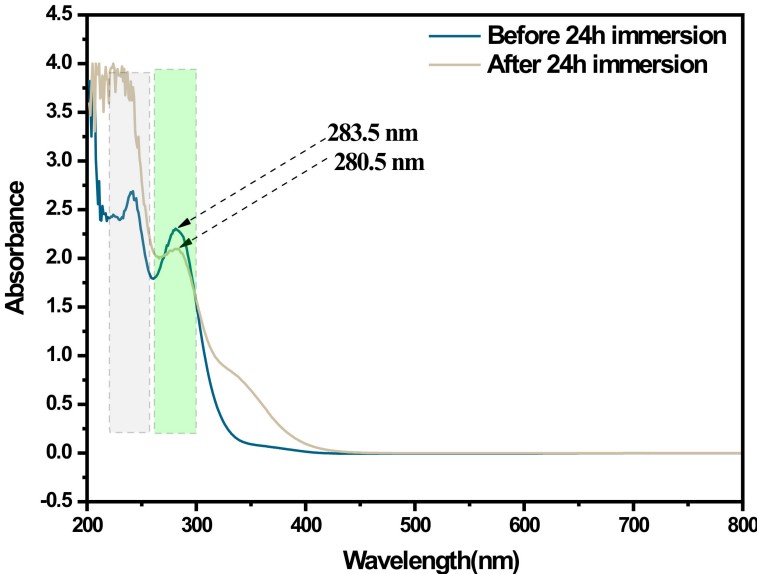

**Figure 7.** UV-visible plots of CHA extract in 1 M HCl medium before and after iron immersion at 298 K.

Before immersion, two intense peaks of wavelengths 283.5 nm and 245 nm are observed. After immersion, two peaks are localized and shifted with respect to the one previously observed, which implies the formation of electrostatic interactions between the ferrous ions and the active cellulose dipoles [54].

### 3.6. SEM Analysis

We used a scanning electron microscope to see the metal surface in order to see how the CHA extract affected it. The SEM pictures of the metal surfaces both with and without the CHA extract are shown in Figure 8. Due to the metal's quick corrosion, we can plainly detect a corroded but also damaged surface after submerging it in an aggressive medium without an inhibitor. In contrast, the smoother surface of the steel upon adding 800 mg/L of CHA toward the HCl solution is a result of the biopolymer's efficient adsorption. As a result, a protective layer forms, altering the steel/electrolyte contact and enhancing the steel's resilience to intense acid assault [55].

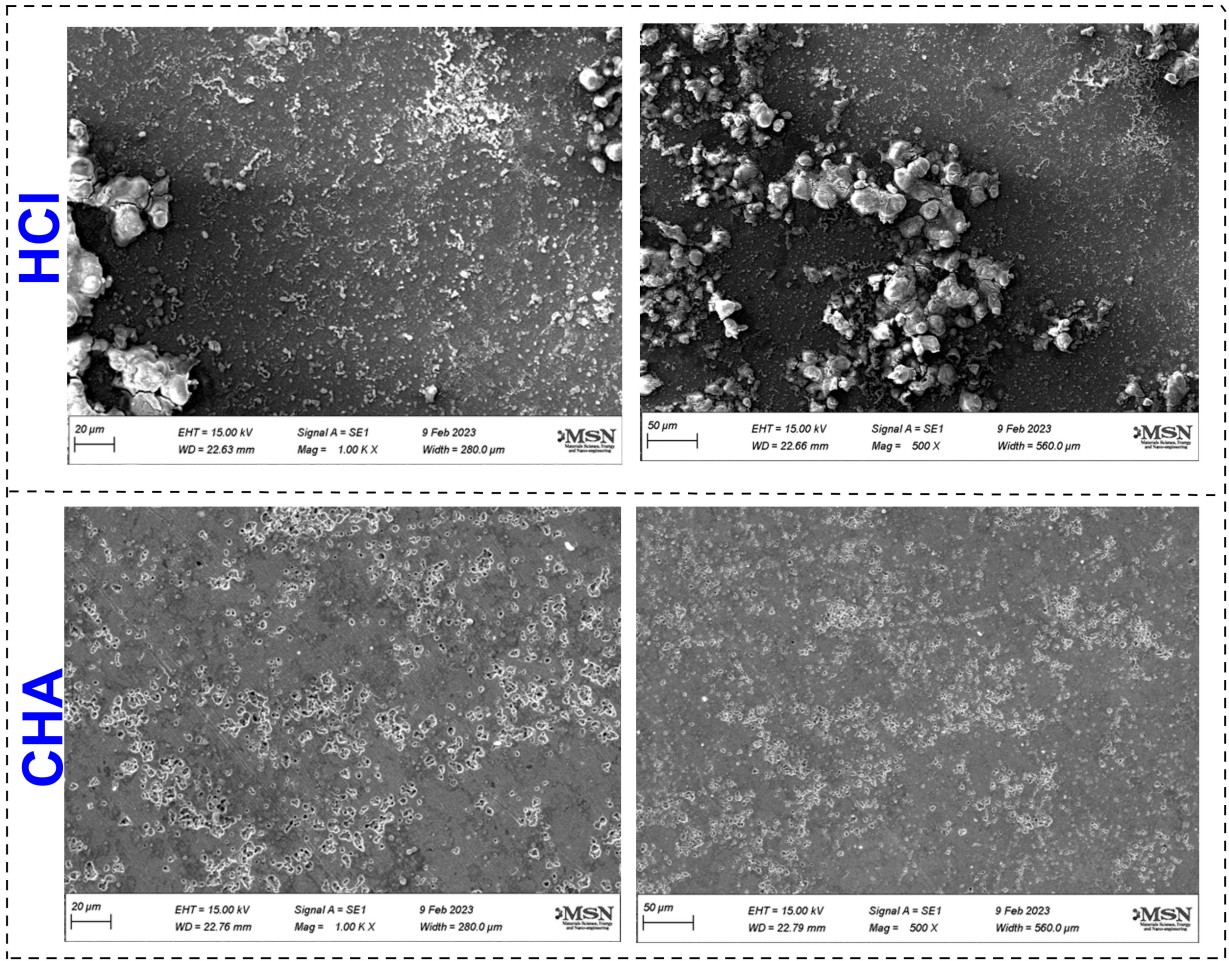

**Figure 8.** SEM images of the mild steel surface in the absence and presence of 800 mg/L of the plant extract CHA *Artemisia herba-alba*.

### 3.7. Theoretical Analysis

#### 3.7.1. DFT Results

To determine the adsorption centers for inhibitors, the Mulliken charge calculation is used. Figure 9 shows the computed Mulliken charges of a few chosen atoms.

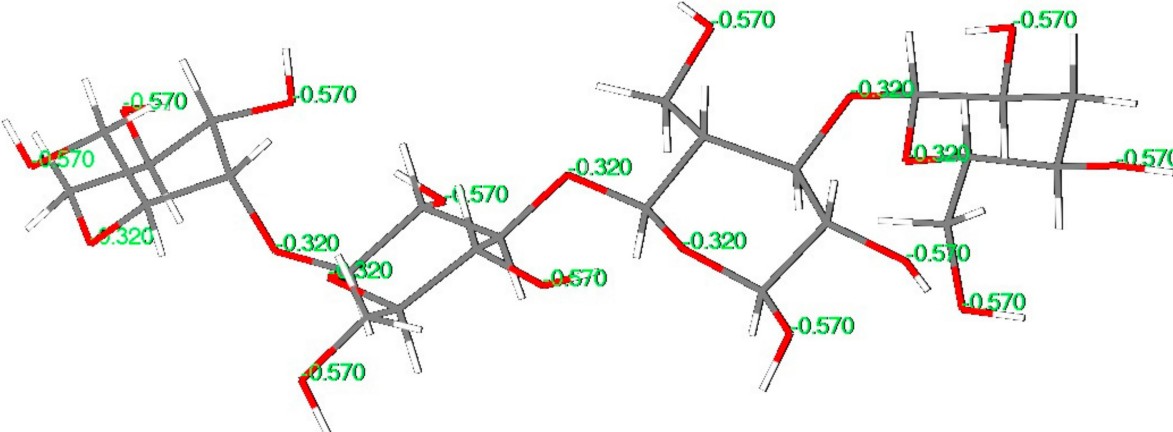

**Figure 9.** Distribution of the Mulliken charge values of O atoms of the biopolymer molecule CHA.

One can see that the heteroatom produces a pronounced surplus of negative charges, thus being able to act as a nucleophilic reagent [56,57]. The Mulliken charges of the atoms, determined for the compound, indicate that the most negative atom is O in red ($-0.570$) and the latter constitutes the active adsorbent center.

Organic chemicals' inhibitory effects often depend on the molecules' ability to adhere to metal surfaces; this adsorption is dependent on the molecules' molecular structures [58]. The DFT has several advantages and appears adequate for generating the electronic data required for inhibitory activity when compared to other quantum chemical approaches for the assessment of corrosion inhibitors [50]. For describing the adsorption selectivity of inhibitors, the electron density distributions of border orbitals are crucial [59]. The HOMO and LUMO, the inhibitor's border molecular orbitals, are thought to be intimately connected to the responsiveness of the inhibitor [60].

The energy of the $\Delta E = (E_{LUMO} - E_{HOMO})$, $E_{LUMO}$, and $E_{HOMO}$ were taken into consideration as quantum-chemical indicators. $E_{HOMO}$ is frequently related to a molecule's capacity for electron donation. With higher $E_{HOMO}$ levels, inhibition becomes more effective. High $E_{HOMO}$ values show that the molecule has the propensity to give electrons to suitable acceptor molecules that have vacant, low-energy atomic orbitals. $E_{LUMO}$ denotes a molecule's capacity to take electrons [61]. The molecule is likely to quickly take electrons from donor molecules, according to the lowest value of $E_{LUMO}$ [44]. Calculated parameters such as $E_{HOMO}$, $E_{LUMO}$, and $\Delta E = E_{LUMO} - E_{HOMO}$, and ionization potential (I), electron affinity (A), electronegativity ($\chi$), overall hardness ($\eta$), overall softness ($\sigma$), and the fraction of electrons transferred ($\Delta N$) are shown in Table 5.

**Table 5.** Calculated theoretical parameters for the biopolymer CHA molecule present in the leaves and stems of *Artemisia herba-alba*.

| Theoretical Parameters | Biopolymer CHA |
|---|---|
| $E_{HOMO}$ (eV) | $-5.812$ |
| $E_{LUMO}$ (eV) | $1.172$ |
| $\Delta E$ ($E_{HOMO} - E_{LUMO}$) (eV) | $6.984$ |
| I (eV) | $5.812$ |
| A (eV) | $-1.172$ |
| $\chi$ (eV) | $2.320$ |
| $\eta$ (eV) | $3.492$ |
| $\sigma$ (eV$^{-1}$) | $0.286$ |
| Chemical potential ($\mu$) | $-2.320$ |
| $\Delta N$ | $0.670$ |
| $\Delta E_{\text{back-donation}}$ | $-0.873$ |

The boundary orbitals of the molecule reveal its chemical reactivity as well as kinetic stability. As adding an electron toward a high-altitude LUMO and removing electrons from a low-altitude HOMO are energetically unfavorable, a large HOMO–LUMO gap predicts great kinetic stability along with low chemical reactivity.

The overall electronegativity ($\chi$) is another important reactivity parameter that represents the electron-attracting ability of the molecule. The stronger the electronegativity, the stronger the ability to attract electrons from the metal, resulting in strong interactions and increased corrosion protection.

According to Table 5, the electronegativity value is appreciable, indicating that this species tends to attract electrons from the metal surface. Additionally, absolute hardness ($\eta$) and softness ($\sigma$) are important properties that measure both the stability and reactivity of inhibitor molecules. A lower hardness value and a higher softness value consist of high chemical reactivity and, therefore, high inhibition efficiency [62].

The inhibition gained via electron donations is described by the fraction of electrons transferred ($\Delta N$). The ability of the metal surface to donate electrons rises with inhibition efficiency [41].

The cellulose inhibitor's optimized molecular structures, MEP, LUMO, and HOMO, are shown in Figure 10; in contrast, Table 5 presents various quantum chemical parameters for the molecule that were calculated using DFT. Figure 10 shows a visual representation of the density distributions of the HOMO but also LUMO frontier orbitals, the electron density surfaces, as well as the electrostatic potential (MEP) of the cellulose inhibitor forms.

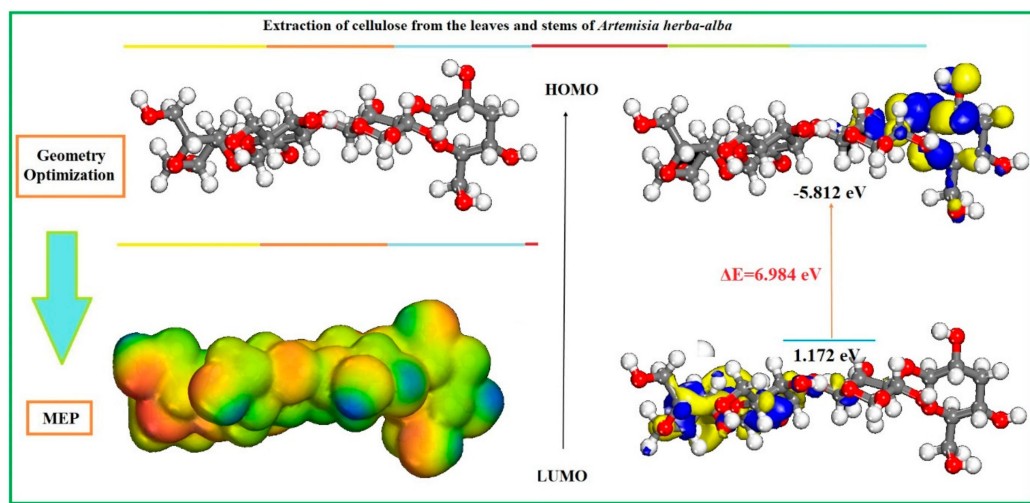

**Figure 10.** Optimized geometry, MEP pictures, and HOMO and LUMO for the cellulose molecule present in the leaves and stems of *Artemisia herba-alba*.

The HOMO and LUMO of the contact determine its capacity to send and receive electrons, respectively. As shown in Figure 10, the electron densities of HOMO for the cellulose inhibitor forms are mainly in the 6-(hydroxymethyl)-2-methoxy-5-methyltetrahydro-2H-pyran-3,4-diol regions bound to oxygen heteroatoms and -OH groups. These sites could donate electrons to the surface of the steel to form coordinate bonds, whereas the electron densities of LUMO mainly focus on the other side on 6-(hydroxymethyl)-5-methoxy-2-methyltetrahydro-2H-pyran-3,4-diol regions bound to oxygen heteroatoms and -OH groups, which means that these atomic centers could be the steel electron acceptor to form bonds.

Cellulose inhibitor reactive sites were verified by the MEP study. MEP is a visual method for understanding electrophilic and nucleophilic attack sites [63]. Nucleophilic and electrophilic centers are shown by a negative (red) and positive (blue) potential, respectively. As shown in Figure 10, the red region is distributed over the oxygen sites, which means

that these sites are the active centers that contribute to the formation of covalent bonds with a mild steel surface.

### 3.7.2. MC and MS Simulations

Both significant interacting systems and the structural analysis of molecules can benefit from the use of MC and MD simulations. These simulations act as dynamic and structural models for comprehending experimental data. Using MC as well as MD simulations, the activity of inhibitors upon this surface was investigated. These methods are helpful and cutting-edge tools for researching how inhibitors interact with metal surfaces [64].

The equilibrium configurations of the biopolymer CHA inhibitor molecules are shown in Figure 11 on the Fe(110) surface. The creation of the nearly parallel orientation on the steel surface is due to the relatively similar distribution of HOMO and LUMO concentrations throughout the molecule (the molecule's interaction with the iron atoms). Figure 12 depicts the computed adsorption energy.

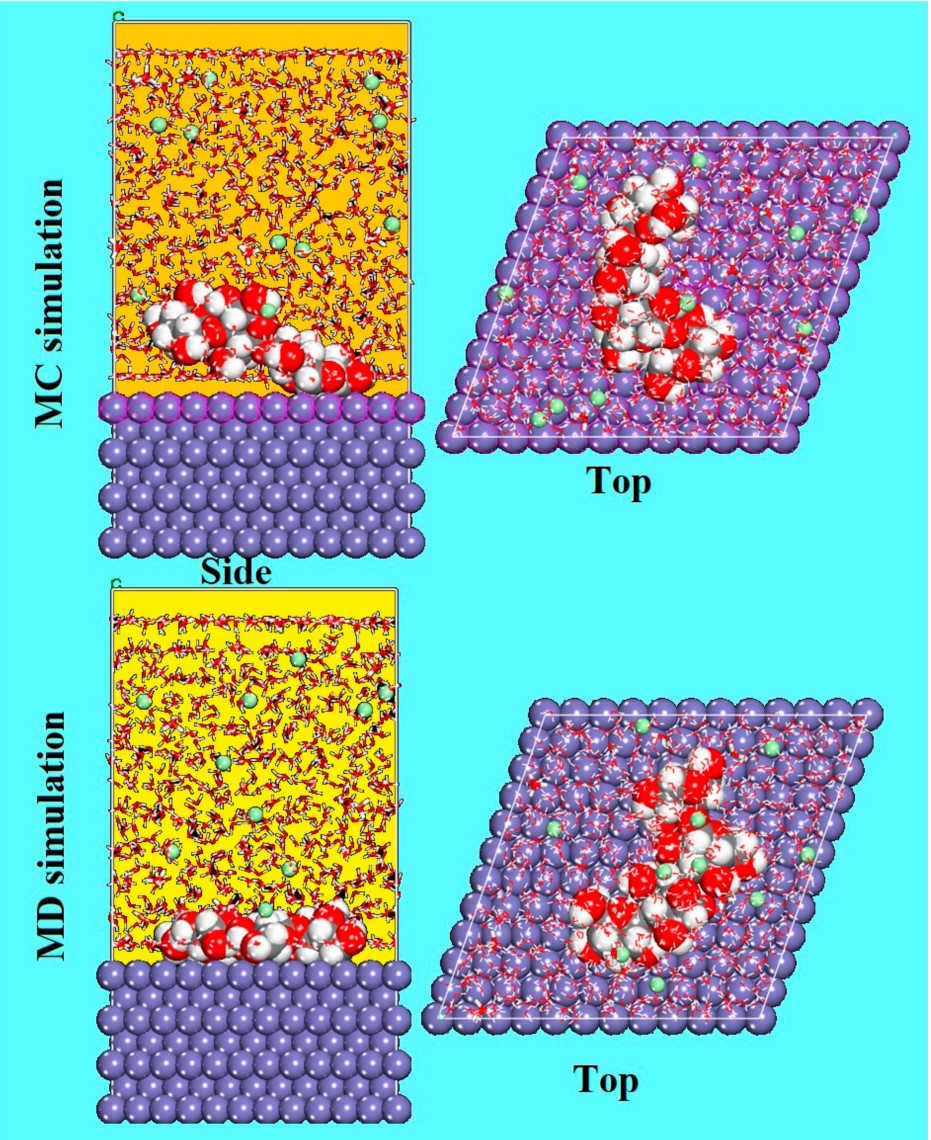

**Figure 11.** MC and MD simulations results from adsorption configurations and positions of the biopolymer CHA molecule present in the leaves and stems of *Artemisia herba-alba*.

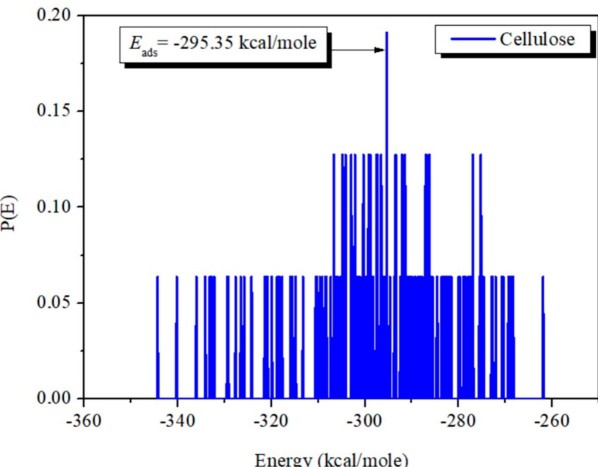

**Figure 12.** Distribution of the $E_{ads}$ of the biopolymer CHA inhibitor via MC simulation.

The strong connection in-between the inhibitor molecule and the Fe surface, as indicated by the substantial negative value of adsorption energy, reveals the spontaneity of the adsorption process [65]. The cellulose molecule's high $E_{ads}$ value ($-295.35$ kcal/mol) demonstrates its higher stability (inhibitor/surface contact), which raises the efficacy of its inhibition. This outcome is consistent with the inhibition efficiencies that were established through experimentation.

The last few years have seen a rapid increase in the use of MD simulations to understand the interaction between the corrosion inhibitor and the metal surface [66]. Thus, in this study, an MD simulation was performed to study the adsorption behavior of the cellulose inhibitor on an Fe(110) surface. The most energetic stable positions of cellulose inhibitors are found by studying the temperature variations in MD simulation analyses. As seen in Figure 13, the temperature drift is minimal, suggesting that the MD of our system was effective [67].

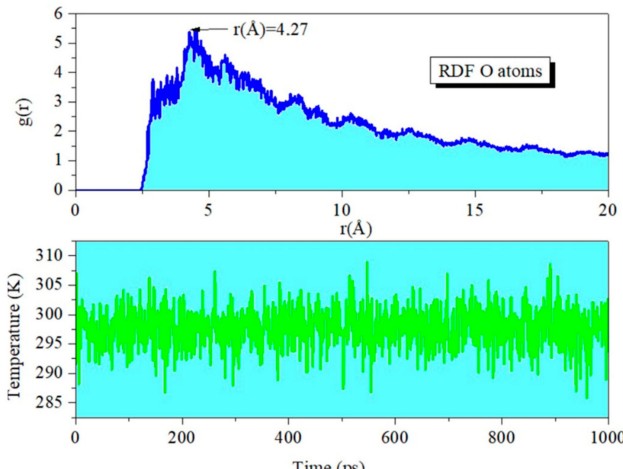

**Figure 13.** The temperature variation (T = 298 K) and the RDF of the O atoms for the biopolymer CHA molecule found in *Artemisia herba-alba* stems and leaves.

The iron-atom bond length with the biopolymer CHA molecules found in *Artemisia herba-leaves alba*'s and stems was measured using RDF analysis. Calculating bond length values allowed us to identify the various sorts of bonds that were produced [65]. The sort of adsorption events resulting on the metal is indicated by peaks in the RDF graph that appear at certain distances from the metal surface [68]. The chemisorption process is considered whenever the peak is present within 1 Å and 3.5 Å; however, for physisorption, the RDF peaks are anticipated to be present at distances larger than 3.5 Å.

Because of the closeness of the biopolymer CHA molecule present in the leaves and stems of *Artemisia herba-alba* atoms to the metal surface, as shown in Figure 13, the derived cellulose molecules have a relatively strong interaction with the surface. This association is what supports the reflected inhibitory performance of the inhibitor [69].

### 3.8. The Inhibition Mechanism

It is widely assumed that the adsorption of an inhibitor at the metal–solution interface is the initial step in the corrosion inhibition process in an aggressive acid solution. The following mechanism is hypothesized for the interaction of inhibitors on the surface of mild steel in an acid solution [3,44]. The uncharged inhibitor molecules can then interact chemically with the metal surface. The inclusion of heteroatoms (O) in the inhibitor molecule with unbound lone pair electrons may facilitate the chemical adsorption of the inhibitor on the metal surface (Figure 14) [70].

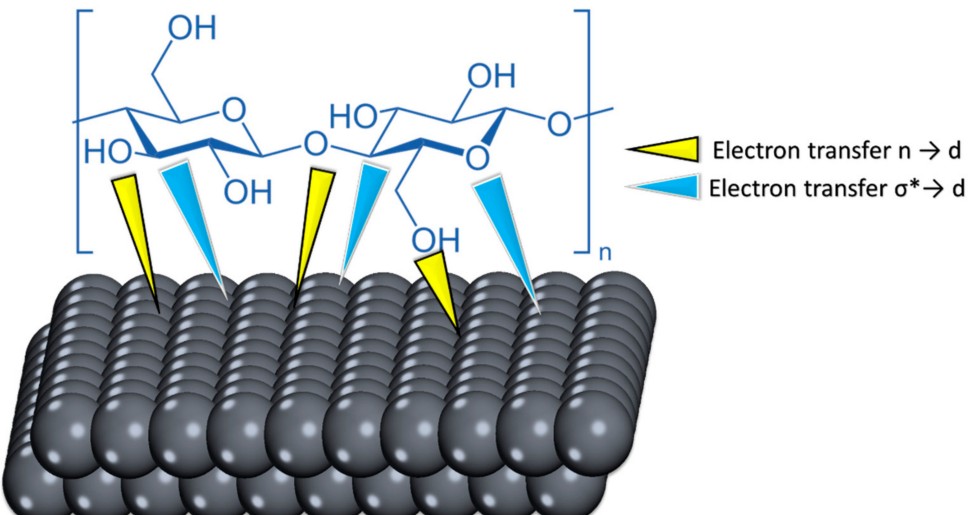

**Figure 14.** Schematic illustration of the adsorption mechanism of the biopolymer CHA onto mild steel.

### 4. Conclusions

This manuscript was devoted to the study of the inhibitory effect of a CHA biopolymer extracted from the plant Artemisia herba-alba, toward the corrosion of mild steel in a 1 M HCl acid medium. The CHA extract is a better inhibitor with an inhibitory efficiency that exceeds 96% at 800 mg/L. The high efficiency of CHA extract can be explained on the basis of its molecular structure. On the other hand, the tested CHA inhibitor adsorbs easily on the metal surface according to the Langmuir isotherm with an adsorption energy of about −20.73 KJ/mol, which shows the presence of chemical and physical bonding between the metal and AMP inhibitor. Analysis of the PDP curves shows that the CHA inhibitor exhibits a mixed characteristic. The inhibitory efficiency of the CHA extract decreases with increasing temperature and the activation energy value Eact increases from 43.20 KJ/mol to 77.79 KJ/mol in the presence of CHA. SEM analysis shows the formation of an organic layer on the mild steel surface in the presence of AMP. The results of the inhibitor quantum chemical calculations provide evidence for the metal–inhibitor interaction and corroborate the experimental results. The DFT calculations reveal that the adsorption centers are located on the O atoms of the inhibitor structure. According to MC calculations, the inhibitors interact strongly with the metal, generating a surface film that blocks the migration of corrosion species to the surface, thereby decreasing the corrosion rate of the metal.

**Author Contributions:** Conceptualization, W.D. and R.H.; methodology, A.E.A., A.O. (Abdelouahad Oussaid) and A.A.; software, A.B.; validation, O.D., K.Z. and R.H.; formal analysis, W.D.; investigation, W.D. and R.H.; resources, A.A.; data curation, W.D.; writing—original draft preparation, R.H.; writing—review and editing, A.O. (Adyl Oussaid).; visualization, F.B. and K.Z.; visualization and supervision, S.-C.K. and E.E.E.; project administration, W.D.; funding acquisition, S.-C.K. All authors have read and agreed to the published version of the manuscript.

**Funding:** This research was funded by the Korean National Research Foundation (NRF), which is funded by the Ministry of Education, grant number 2020R1I1A3052258.

**Institutional Review Board Statement:** Not applicable.

**Informed Consent Statement:** Not applicable.

**Data Availability Statement:** The data presented in this study are available on request from the corresponding author.

**Conflicts of Interest:** The authors declare no conflict of interest.

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
