# Peer review of "Anti-Corrosion Coating Formation by a Biopolymeric Extract of Artemisia herba-alba Plant: Experimental and Theoretical Investigations"

_coatings, doi:10.3390/coatings13030611_

Round 1
Reviewer 1 Report
The authors of the manuscript “Anti-corrosion coating formation by a biopolymeric extract of Artemisia herba-alba plant: Experimental and theoretical investigations” mention that the novelty of this research work is to extract a bio-polymer "cellulose" (CAH) from the waste of the Artemisia herba-alba plant and used it as a mild steel corrosion inhibitor in the chemical engineering industry.
Unfortunately, the manuscript is very confusing due to the fact that authors didn’t have a clearly working plan. They use different terms like “the extract of the plant Artemisia herba-alba”, “the CHA extract”, …without continuity. I suggest to be more clear in defining the experimental samples that was tested by using a table with their description.
Also, is not specified the relevance of the fact that cellulose was extracted from the stems or from the leaves of the plant Artemisia herba-alba. I think that is not relevant for the current manuscript.
Reformulate the sentences like “To characterize the iron-bio-polymer complex, UV-visible spectrometry tests are performed.”, “the effect of CHA bio-polymer concentration on the surface of mild steel MS”.
Add more clarification to the “Figure 2: The evolution of inhibitory efficiency and corrosion rate in 1M HCl without and with Artemisia herba-alba extract.”
Also, the legend for figure 8 must be adapted to the 3 images used.
Regarding the SEM analysis, not “3.6. SEM visualization”, I suggest inserting more figures. Minimally 2 must be shown, in order to prove the homogeneity of the coating with the CHA extract and another one to prove the protection layer (a cross-section analysis to see the protection layer is recommended).
The authors say in conclusion that the manuscript was devoted to the study of the inhibiting effect of a bio-polymer extracted from the plant Artemisia herba-alba, vis-à-vis the corrosion of mild steel in an acid medium HCl 1M. Even the authors try to follow the effects of temperature on this aspects, I think that more important is to follow the effect in time. In the case of chemical engineering industry, the stability of the inhibitors against corrosion is very important.
Also, the authors say “The CHA extract is a better inhibitor with an inhibitory efficiency that exceeds 96% at 800 mg/l.”. Please, describe this performance in comparation with other inhibitors from the literature.
Reformulate the sentence “Which gives a lower resolution of the steel and formation of a protective barrier film justified by the SEM technique.”
The structure of the manuscript must be correlated with the aim.
The conclusion is not sustained by the experimental results, at this moment.
Author Response
- Reviewer #1
The authors of the manuscript “Anti-corrosion coating formation by a biopolymeric extract of Artemisia herba-alba plant: Experimental and theoretical investigations” mention that the novelty of this research work is to extract a bio-polymer "cellulose" (CAH) from the waste of the Artemisia herba-alba plant and used it as a mild steel corrosion inhibitor in the chemical engineering industry.
- Unfortunately, the manuscript is very confusing due to the fact that authors didn’t have a clearly working plan. They use different terms like “the extract of the plant Artemisia herba-alba”, “the CHA extract”, …without continuity. I suggest to be more clear in defining the experimental samples that was tested by using a table with their description.
Response: Dear Reviewer, Thank you for your careful review of our manuscript. In this work, we extracted a CHA biopolymer from the local plant Artemisia herba-alba, whose objective is to use it as a corrosion inhibitor of mild steel in 1M HCl medium. I have tried to be clearer in defining the CHA extract in the manuscript. Please check the manuscript.
- Also, is not specified the relevance of the fact that cellulose was extracted from the stems or from the leaves of the plant Artemisia herba-alba. I think that is not relevant for the current manuscript.
Response: Dear Reviewer, Thank you for your remark. The extracted CHA biopolymer (cellulose) was extracted by both (stems and leaves). But, the objective of this work is to use the CHA biopolymer extracted by the leaves as a corrosion inhibitor of mild steel in the acidic environment.
- Reformulate the sentences like “To characterize the iron-bio-polymer complex, UV-visible spectrometry tests are performed.”, “the effect of CHA bio-polymer concentration on the surface of mild steel MS”.
Response: Dear Reviewer, Thank you for your careful review of our manuscript. We have rephrased both sentences. Please check the manuscript.
- Add more clarification to the “Figure 2: The evolution of inhibitory efficiency and corrosion rate in 1M HCl without and with Artemisia herba-alba extract.”
Response: Dear Reviewer, Thank you for your careful review of our manuscript. We have added more details in figure 2. Please check the manuscript.
- Also, the legend for figure 8 must be adapted to the 3 images used.
Response: Dear reviewer, Thank you for your comment. We have tried to clarify figure 8. Please check the manuscript.
- Regarding the SEM analysis, not “3.6. SEM visualization”, I suggest inserting more figures. Minimally 2 must be shown, in order to prove the homogeneity of the coating with the CHA extract and another one to prove the protection layer (a cross-section analysis to see the protection layer is recommended).
Response: Dear Reviewer, Thank you for your careful review of our manuscript. We tried to make two images for each case with different magnifications to show the formation of the polymeric layer on the metal surface. In addition, we have taken into consideration making a cross-sectional analysis in the next works. Please check the manuscript.
- The authors say in conclusion that the manuscript was devoted to the study of the inhibiting effect of a bio-polymer extracted from the plant Artemisia herba-alba, vis-à-vis the corrosion of mild steel in an acid medium HCl 1M. Even the authors try to follow the effects of temperature on this aspects, I think that more important is to follow the effect in time. In the case of chemical engineering industry, the stability of the inhibitors against corrosion is very important.
Response: Dear Reviewer, Thank you for your careful review of our manuscript. In this work, we have followed the effect of temperature at a fixed time of one hour. For the effect of time with the change of temperature, it is a very important remark; we can do it in the next work.
- Also, the authors say “The CHA extract is a better inhibitor with an inhibitory efficiency that exceeds 96% at 800 mg/l.”. Please, describe this performance in comparation with other inhibitors from the literature.
Response: Dear Reviewer, Thank you for your careful review of our manuscript. We have tried to perform a comparative study with other extracts. Please check the manuscript.
- Reformulate the sentence “Which gives a lower resolution of the steel and formation of a protective barrier film justified by the SEM technique.”
Response: Dear reviewer, Thank you for your careful review of our manuscript. We have reformulated this sentence. Please check the manuscript.
- The structure of the manuscript must be correlated with the aim.
Response: Dear reviewer, Thank you for your careful review of our manuscript. We have tried to explain the objective of this work in the abstract, introduction and conclusion sections
- The conclusion is not sustained by the experimental results, at this moment.
Response: Dear reviewer, Thank you for your careful review of our manuscript. We have tried to add the maximum of the experimental results obtained in the conclusion section. Please check the manuscript.
Reviewer 2 Report
In this work, authors utilized the green plant extract to effectively retard the corrosion of mild steel in HCl medium. Rich data has been collected to verify the main conclusions. In my opinion, this paper can be published in Coatings after properly dealing with the major issues as follows.
-- Artemisia herba-alba extract (CAH) was used as an efficient corrosion inhibitor for mild steel in 1 M HCl solution. It is admitted that CAH, indeed, exerts the favorable protection efficiency (max. 96.17%) from the experimental results. However, similar studies have reported the extract from the same kind of plant as a potential green corrosion inhibitor that can be searched. Thereby, authors must clearly express the key advance of this presented work, which directly decides whether this report can be published or not.
-- CAH was defined as a polymeric inhibitor that can be dissolved in the aqueous solution; and FT-IR technique was deployed to confirm the structure proposed in Scheme 1 (bottom). However, only FT-IR spectra were insufficient to support this clear molecular structure. As for polymers, besides the general formula, conformation is the other key point to present their spatial structure. I can see in Scheme 1 that geometric isomerism is shown for CAH. How to verify it? As a friendly suggestion, authors could introduce the spectrum for standard CAH compound, make a comparison and thus confirm its structure.
-- Several typical frequencies should be marked in EIS spectra to make a comparison among different concentrations of CAH. Another, the scale of real and image domains should be the same.
-- What is the regression coefficient of Langmuir fitting in Figure 6. I think it is not very satisfied. In addition, the electron-hoping categories should be mark in UV-vis spectra. From the UV-vis results, the interaction between inhibitor molecules and corroded species can be revealed clearly (e.g., Sustain. Chem. Pharm., 2022, 29: 100821), which has attracted the increasing attention in relevant studies. Consequently, authors should mention it and make a discussion.
-- DFT calculations also have some flaws. On one hand, reactive parameters of a sole corrosion inhibitor are meaningless; and the first-principle calculations should be focused on the adsorption model. On the other hand, dominant solvent model has been developed as an optimal choice to obtain the related items for the corrosion inhibitor. Thus, authors must give a clear statement to definite the limitation of the presented method.
-- This work has a too-heavy self-citation that I can understand. It is proper to present several typical studies of this group.
Author Response
Reviewer #2
In this work, authors utilized the green plant extract to effectively retard the corrosion of mild steel in HCl medium. Rich data has been collected to verify the main conclusions. In my opinion, this paper can be published in Coatings after properly dealing with the major issues as follows.
- Artemisia herba-alba extract (CAH) was used as an efficient corrosion inhibitor for mild steel in 1 M HCl solution. It is admitted that CAH, indeed, exerts the favorable protection efficiency (max. 96.17%) from the experimental results. However, similar studies have reported the extract from the same kind of plant as a potential green corrosion inhibitor that can be searched. Thereby, authors must clearly express the key advance of this presented work, which directly decides whether this report can be published or not.
Response: Dear reviewer, Thank you for your careful review of our manuscript. As mentioned in the introductory section, many researchers have used either essential oils or alcoholic extracts that give inhibitory efficiencies that do not exceed 90%. Therefore, the novelty of this work is to extract a new local biopolymer (cellulose), which shows a high inhibitory efficiency of the order of 96.17% at 800 mg/l.
- CAH was defined as a polymeric inhibitor that can be dissolved in the aqueous solution; and FT-IR technique was deployed to confirm the structure proposed in Scheme 1 (bottom). However, only FT-IR spectra were insufficient to support this clear molecular structure. As for polymers, besides the general formula, conformation is the other key point to present their spatial structure. I can see in Scheme 1 that geometric isomerism is shown for CAH. How to verify it? As a friendly suggestion, authors could introduce the spectrum for standard CAH compound, make a comparison and thus confirm its structure.
Response: Dear reviewer, Thank you for your careful review of our manuscript. Really, the FT-IR technique is insufficient to identify the extracted polymer. At this stage, we made a comparative analysis between the extracted polymer and the commercial cellulose (Scheme 1), in order to confirm the structure of extracted cellulose.
- Several typical frequencies should be marked in EIS spectra to make a comparison among different concentrations of CAH. Another, the scale of real and image domains should be the same.
Response: Dear reviewer, Thank you for your careful review of our manuscript. We have added some typical frequencies in the EIS spectrum. Moreover, the scale of the real and image domains is the same. Please check the manuscript.
- What is the regression coefficient of Langmuir fitting in Figure 6. I think it is not very satisfied. In addition, the electron-hoping categories should be mark in UV-vis spectra. From the UV-vis results, the interaction between inhibitor molecules and corroded species can be revealed clearly (e.g., Sustain. Chem. Pharm., 2022, 29: 100821), which has attracted the increasing attention in relevant studies. Consequently, authors should mention it and make a discussion.
Response: Dear reviewer, Thank you for your careful review of our manuscript. we tested the other isotherms, but we found that the adsorption of CHA on the metal is obeyed by the Langmuir isotherm whose correlation coefficient is about 0.966, which is close to unity. On the other hand, as we mentioned in Figure 7, there is a shift in the wavelength level when immersing the metal, these results clearly show the formation of an Iron-CHA complex. Thanks.
- DFT calculations also have some flaws. On one hand, reactive parameters of a sole corrosion inhibitor are meaningless; and the first-principle calculations should be focused on the adsorption model. On the other hand, dominant solvent model has been developed as an optimal choice to obtain the related items for the corrosion inhibitor. Thus, authors must give a clear statement to definite the limitation of the presented method.
Response: Numerous studies in the literature have demonstrated that DFT estimates of inhibitors are critical in evaluating their absorptive characteristics and adsorption sites. DFT-derived parameters, in particular, have been shown to be essential in evaluating intrinsic inhibitor characteristics, which give them anti-corrosive properties [for example: D. Iravani et al., Colloids Surfaces A Physicochem. Eng. Asp. 656, 130544 (2023); A. Farhadian et al, Energy 269, 126797 (2023); R. Ganjoo et al. Mol. 2023, Vol. 28, Page 1581 28, 1581 (2023); M. Alahiane et al., Appl. Surf. Sci. 612, 155755 (2023); C. Tang et al., ACS Sustain. Chem. Eng. (2022); M. Omidvar et al., Energy and Fuels (2022)]. As a result, in accordance with those studies, we have computed the most prevalent of such parameters here. Although we concur that evaluating the electronic properties of inhibitors on metal surfaces using DFT provides a more comprehensive view, due to computational constraints, such calculations were not feasible in this research. We agree with the reviewer's statement and will do our best to include those calculations in our future work.
- This work has a too-heavy self-citation that I can understand. It is proper to present several typical studies of this group.
Response: Response: Dear reviewer, Thank you for your careful review of our manuscript. We have deleted some self-citation.
Round 2
Reviewer 2 Report
Due to the proper revision, I recommend it publication.